# Effects of Work Stress and *Period3* Gene Polymorphism and Their Interaction on Sleep Quality of Non-Manual Workers in Xinjiang, China: A Cross-Sectional Study

**DOI:** 10.3390/ijerph19116843

**Published:** 2022-06-03

**Authors:** Juan Wang, Jiwen Liu, Huiling Xie, Xiaoyan Gao

**Affiliations:** 1Xinjiang Key Laboratory of Special Environment and Health Research, Urumqi 830011, China; 13139638739@163.com; 2Department of Public Health, Xinjiang Medical University, Urumqi 830011, China; liujiwendr@163.com (J.L.); sherilyn@tom.com (H.X.)

**Keywords:** work stress, sleep quality, *Per3* gene polymorphism, non-manual workers

## Abstract

Work stress has been found to be associated with sleep quality in various occupational groups, and genetic factors such as variable number tandem repeat polymorphism in the *Period3* (*Per3*) gene also influence the circadian sleep-wake process. Therefore, the present study aimed to evaluate the sleep quality status of non-manual workers in Xinjiang, China and to analyse the effects of work stress and *Per3* gene polymorphism and their interaction on sleep quality. A cluster sampling method was used to randomly select 1700 non-manual workers in Urumqi, Xinjiang. The work stress and sleep quality of these workers were evaluated using the Effort–Reward Imbalance Inventory (ERI) and the Pittsburgh Sleep Quality Index (PSQI). Next, 20% of the questionnaire respondents were randomly selected for genetic polymorphism analysis. The polymerase chain reaction-restriction fragment length polymorphism technique was used to determine *Per3* gene polymorphism. The detection rate of sleep quality problems differed between the different work stress groups (*p* < 0.05), suggesting that non-manual workers with high levels of work stress are more likely to have sleep quality problems. Regression analysis revealed that the *Per3* gene (*OR* = 3.315, 95% *CI*: 1.672–6.574) was the influencing factor for poor sleep quality after adjusting for confounding factors, such as occupation, length of service, education, and monthly income. Interaction analysis showed that Per3^4/5^^,5/5^ × high work stress (*OR* = 2.511, 95% *CI*: 1.635–3.855) had a higher risk of developing sleep quality problems as compared to Per3^4/4^ × low work stress after adjusting for confounding factors. The structural equation modelling showed no mediating effect between work stress and *Per3* gene polymorphism. The results of this study show that both work stress and *Per3* gene polymorphism independently affect sleep quality of nonmanual workers from Xinjiang, and the interaction between these two factors may increase the risk of sleep quality problems. Therefore, to improve sleep quality, individuals with genetic susceptibility should avoid or reduce as much as possible self-stimulation by work-related exposures such as high levels of external work stress.

## 1. Introduction

Sleep is an integral part of maintaining the physiological functions of the body and is essential for our physical and mental health. Good sleep quality can reduce the risk of adverse diseases such as cardiovascular diseases, diabetes, mental and cognitive disorders, and accidental injuries [1]. Poor sleep quality is usually characterized by difficulties in falling asleep and maintaining sleep [2], which can easily lead to significant discomfort and thus affect daily life. The prevalence of sleep quality problems in occupational groups varies by region, with 58.2% of employees over the age of 18 years in the United States reporting sleep problems in the past month [3], and 30–45% and 40.2% of corporate employees and civil servants in Japan reporting sleep-related issues, respectively [4,5]. Sleep quality problems are becoming a major concern in today’s society, and the chronic occurrence of sleep problems increases the risk of many diseases and places a heavy burden on health services worldwide [6].

Sleep quality problems in the occupational population can lead to increased absenteeism and reduced productivity, which can affect the ability to work [7] and can even lead to workplace accidents, thereby putting workers’ own lives at risk [8]. Non-manual workers are a specific type of occupational group whose physical and mental health, literacy, and quality of life can greatly affect economic and social development. With the rapid development of technology, the proportion of work based on mental ability is gradually increasing, which has minimised the need to perform manual work and thus has greatly improved work efficiency and industrial performance. However, fierce business competition in the industry has also overloaded the workforce with a heavy work schedule, thereby creating an imbalance between work capacity, coping resources, and demand, which can affect physical and mental health and consequently influence sleep quality.

Factors affecting sleep quality are multifaceted. In addition to demographic characteristics such as gender and age, work-related factors such as job stress and effort–reward imbalance may also affect sleep quality [9,10]. In early studies, most researchers in China and abroad focused on the effects of environmental exposure factors on sleep quality in occupational populations, and work stress, as one of the exposure factors, is a common cause of sleep difficulties. Work stress, also known as work stress and occupational strain, is the physical and psychological response of an individual due to a perceived incongruity between the demands and conditions of work and his or her own abilities and resources [11]. Many studies have found an association between work stress and insomnia, sleep disturbance, and poor sleep quality [12,13,14]. A non-randomised pilot study of adults concluded that prolonged exposure to stress with high work demands and low work rewards could affect sleep quality [15]. Stressors resulting from the imbalance between high work demands and low rewards were found to be associated with shorter sleep duration and insomnia [16,17]. Work stress can affect sleep quality by acting on the body through factors such as stress response and stress coping. When stress is excessive and lasts for too long, it can cause an imbalance in the body’s neuroendocrine regulation, with the occurrence of adverse reactions such as anxiety and depression [18], ultimately leading to sleep problems. All these results suggest a close link between work stress and sleep quality.

With the rapid development of molecular biology, studies have shown that both genetic and environmental factors affect sleep quality [19]. The system that regulates biological rhythms in the body is called the biological clock, which plays a leading role in regulating circadian rhythms. The *Period3* (*Per3*) gene, one of the core genes of the biological clock, is located on chromosome 1p36, with a total length of 60,475 bp and 21 exons and having four or five tandem repeats (variable number tandem polymorphism [VNTP]) in exon 18. This region has a series of predicted phosphorylation sites and is polymorphically expressed in the population. Various studies have reported that *Per3* gene polymorphism is strongly associated with sleep physiology and individual circadian preferences [20,21]. However, some studies have reported no evidence of an association between *Per3* gene polymorphism and sleep quality and daytime preferences [22]. In a Japanese study, a variable number tandem repeat (VNTR) genotyping of the *Per3* gene in subjects revealed no significant correlation between VNTR and ‘morning and evening’ preferences [23]. Therefore, the relationship between *Per3* gene polymorphism and sleep quality needs to be further investigated.

Previous studies have reported the relationship between work stress, *Per3* gene polymorphism, and sleep quality; however, fewer studies have examined the effects of work stress, *Per3* gene polymorphism, and their interaction on sleep quality in non-manual workers. Therefore, the present study was conducted to analyse the effects of work stress and *Per3* gene polymorphism on sleep quality of non-manual workers by combining epidemiological surveys and molecular biology experiments to further explore the effects of the interaction of these two factors on sleep quality and provide a theoretical basis for better improving the sleep quality of non-manual workers and improving their physical and mental health and quality of life.

## 2. Materials and Methods

### 2.1. Subjects

This study was conducted in Urumqi, Xinjiang, China, and the survey period was from January 2021 to December 2021. The study protocol was approved by the Ethics Committee of Xinjiang Medical University, and all participants voluntarily completed a written informed consent form prior to the survey. Occupational groups in the first and second major categories ((administrative state organizations, party and mass organizations, enterprises, institutions, professional, and technical personnel, etc.) were selected as the target population according to the Dictionary of Occupational Classification of the People’s Republic of China. A cluster sampling method was used to select non-manual workers in Urumqi, Xinjiang for the questionnaire survey.

Inclusion and exclusion criteria: Active workers aged 20–60 years with ≥1 year of service who were willing to cooperate with the completion of the questionnaire and the collection of blood samples were included in the study. Workers with a history of traumatic brain injury, hypertension, coronary heart disease, diabetes mellitus, thyroid disease, asthma, chronic bronchitis, and tumours that may cause sleep disorders; a history of other psychiatric disorders (schizophrenia, depression, mania, etc.) that may cause sleep disorders; and having other serious physical illnesses, sleep apnoea syndrome, episodic sleeping sickness, restless leg syndrome, etc. were excluded based on the past history survey. Workers who had been treated for their sleep quality problems with medication or hospitalisation within the last 3 months were also excluded.

Before administering the questionnaire, the investigator questioned the respondents according to the inclusion and exclusion criteria and administered the questionnaire to those who met the inclusion criteria and did not meet the exclusion criteria. Finally, a total of 1700 questionnaires were distributed; after excluding incomplete questionnaires and those with less than 80% of the content filled, the number of valid questionnaires was 1458, with an effective response rate of 85.76%. Twenty percent of the questionnaire respondents; i.e., 292 participants, were randomly selected for the polymorphism detection experiment. After eliminating samples with substandard concentration and purity of the extracted DNA, a total of 251 samples were tested for *Per3* gene polymorphism, and the 251 participants were matched 1:1 using gender and age as the matching factors. A total of 113 pairs were successfully matched, resulting in a case-control study of 226 participants.

### 2.2. Methods

#### 2.2.1. Assessment of Work Stress

A self-administered Effort–Reward Imbalance Inventory (ERI) based on the Effort–Reward Imbalance model was developed by Siegrist in Germany [24]. Currently, the ERI model is the dominant model for assessing work stress in many countries and is used to describe the relationship between workplace characteristics and individual psychological well-being [25,26]. The questionnaire consists of three dimensions: effort, reward, and overcommitment, with 23 items. Of these, the first six items rate ‘effort’, the middle 11 items rate ‘reward’ and the last six items rate ‘overcommitment’. In addition, the first 17 items are scored on a 5-point scale and the last 6 items are scored on a 4-point scale. The effort–reward imbalance index is calculated as: the score of effort/(the score of reward × C), where C is the number of effort items over the number of reward items, i.e., 6/11. If ERI > 1, the person is a high effort-low reward person (high work stress); if ERI ≤ 1, the person is a low effort-high reward person (low work stress).

#### 2.2.2. Assessment of Sleep Quality

The Pittsburgh Sleep Quality Index (PSQI), developed by Dr Buyee in 1989 [27], was used to assess sleep quality. This scale is currently more commonly used in psychiatric clinical assessment abroad because of its ease of use in terms of survey methods and its high correlation with the results of polysomnographic electroencephalography (EEG) tests [28]. The scale is used to evaluate the quality of sleep in the previous month and consists of 19 self-rated and 5 other-rated entries (not involved in scoring), which include 7 sub-items of subjective sleep quality, sleep latency, sleep duration, sleep efficiency, sleep disturbance, hypnotic use, and daytime dysfunction. Each entry is scored on a scale of 0 to 3, and the cumulative score is the total PSQI score, which ranges from 0 to 21, with higher scores indicating poorer sleep quality. In accordance with international standards, the threshold for defining a sleep quality problem is 5. In this study, a total PSQI score of ≥5 was defined as poor sleep quality.

#### 2.2.3. Genotyping

Genomic DNA (gDNA) was extracted using the Blood Genomic DNA Extraction System (non-centrifugal column type) kit (Tiangen Biotech, Beijing, China). *Per3* gene polymorphism analysis was performed using the polymerase chain reaction–restriction fragment length polymorphism (PCR-RFLP) technique. The total volume of each reaction mixture was 25 µL, and gDNA was amplified using PCR instruments (MyCycler, Bio-Rad, Hercules, CA, USA). The PCR reaction conditions were as follows: pre-denaturation at 94 °C for 5 min, followed by denaturation at 94 °C for 40 s, annealing at 58 °C for 30 s, extension at 72 °C for 40 s for a total of 30 cycles, followed by final extension at 72 °C for 12 min. The PCR amplification products were detected by 1.5% agarose gel electrophoresis and observed on a gel imager. The primers and genotypes for the *Per3* gene are listed in Table 1 and Table 2, respectively.

#### 2.2.4. Quality Control

Before the survey was formally conducted, the investigators were professionally trained to learn the content of the questionnaire, the survey language, and the survey method. During the survey, the surveyor clarified the purpose, content, and significance of the survey to the respondents and explained in detail the requirements for completing the questionnaire; the surveyor also attempted to seek the cooperation of the respondents and respected their wishes as far as possible during the survey. The questionnaires were centrally distributed and collected on the spot. The questionnaires were reviewed, and those that were less than 80% completed were excluded and those that passed the survey were numbered. For the polymorphism detection experiment, the researchers wore a white laboratory apron, mask, and gloves before entering the laboratory. The researchers received safety training from the laboratory supervisor in advance of the experiment and familiarised themselves with the laboratory equipment and its operating procedures. During the experiments, the researchers were required to strictly observe the instructions of the laboratory supervisor and the safety regulations of the laboratory. The blood samples were carefully marked; the reagents were prepared in strict accordance with the instructions, and the names of the reagents were checked carefully. To avoid cross-contamination, reagents and samples were stored properly.

#### 2.2.5. Statistical Analysis

Data entry was performed using Epidata 3.1 (The Epidata Association, Odense, Denmark), and data analysis was performed using the statistical software SPSS 26.0 (IBM, Armonk, NY, USA). Comparisons of sleep quality problem detection rates and Hardy–Weinberg genetic equilibrium tests for the *Per3* gene and comparisons of genotypes between the sleep quality groups were performed using chi-square tests. The PSQI scores were statistically expressed as $\bar{x}\pm s$, and two independent samples *t*-test was used for comparison between the groups. Multiple regression analysis was used to determine the effect of the interaction between work stress and *Per3* gene polymorphism on sleep. Pathway analysis between gene–environment–sleep quality was performed using SPSS Amos 24.0 (IBM). Subjects for gene polymorphism analysis were matched using propensity score matching (PSM) with a matching error of 0.02. The significance level was taken as α = 0.05 (two-sided).

## 3. Results

### 3.1. Detection of Sleep Quality Problems among Non-Manual Workers with Different Demographic Characteristics

The results of the survey showed that 1038 of 1458 non-manual workers had sleep quality problems, with a detection rate of 71.19%. The detection rate of sleep quality problems among non-manual workers differed between the smoking groups, but the difference in the detection rate was not statistically significant for the remaining demographic characteristics (*p* > 0.05). The detection rate of sleep quality problems was higher in non-manual workers who smoked (77.5%) than in non-smokers (69.9%) (Table 3).

### 3.2. Association between Work Stress and Sleep Quality

The distribution of sleep quality among non-manual workers differed between the work stress groups, and the differences were statistically significant (*p* < 0.05). The number of non-manual workers with sleep quality problems was significantly higher in the high work stress group than in the low work stress group. This finding suggests that non-manual workers with high levels of work stress are more likely to have sleep quality problems (Table 4).

### 3.3. Association between Work Stress and the Dimensions of Sleep Quality and PSQI Score

The Pittsburgh Sleep Quality Index (PSQI) contains subjective sleep quality, sleep latency, sleep duration, sleep efficiency, sleep disturbance, hypnotic use, and daytime dysfunction, and then the final sum of all dimensions is the total PSQI score. The PSQI scores on all dimensions differed between the different work stress groups, except for the sleep efficiency score, and the differences were statistically significant (*p* < 0.05). All PSQI scores in the high work stress group were higher than those in the low work stress group, thus suggesting that high work stress is associated with poorer sleep quality (Table 5).

### 3.4. Hardy–Weinberg Genetic Equilibrium Test

The Hardy–Weinberg genetic equilibrium test was used to analyse the distribution of genotypes at the 54 bp-VNTR locus of the *Per3* gene. The results showed that the actual and expected values of each genotype in this study were in good agreement, with no statistically significant differences (*p* > 0.05); this finding suggests that the gene frequencies of the 226 subjects were in genetic equilibrium in accordance with the law of genetic equilibrium (Table 6).

### 3.5. Distribution of Sleep Quality across Genotypes and Alleles of the Per3 Gene

The differences in the distribution of sleep quality among genotypes and alleles at the 54bp-VNTR locus of the *Per3* gene were statistically significant. (*p* < 0.05) (Table 7).

### 3.6. Logistic Regression Analysis of Work Stress, Per3 Gene, and Poor Sleep Quality

Of the 226 experimental study subjects, the sleep quality status (Poor sleep quality and non-poor sleep quality) served as the dependent variable, while work stress and *Per3* gene genotypes were considered to be independent variables. Genotype 4/5 and genotype 5/5 were grouped together due to the low number of genotypes 5/5. Multivariate logistic regression analysis was performed. The results showed that both work stress (*OR* = 3.088, 95% *CI*: 1.639–5.820) and *Per3* gene (*OR* = 3.315, 95% *CI*: 1.672–6.574) were influencing factors for poor sleep quality after adjusting for confounding factors, such as occupation, length of service, education, and monthly income (Table 8).

### 3.7. Interaction of Work Stress and Per3 Gene Polymorphism on Sleep Quality in Non-Manual Workers

#### 3.7.1. Logistic Regression Analysis of the Interaction between Work Stress and Per3 Gene Polymorphism on Sleep Quality

The interaction terms for work stress and Per3 gene genotype were further introduced into the multiple regression model. The results showed that, after adjusting for confounding factors, such as occupation, length of service, education, and monthly income, Per3^4/5^^, 5/5^ × high work stress (OR = 2.511, 95% CI: 1.635–3.855) had a higher risk of developing sleep quality problems as compared to Per3^4/4^ × low work stress. This finding suggested that there is an interaction between work stress and Per3 gene polymorphism, and this interaction increases the risk of developing sleep quality problems (Table 9).

#### 3.7.2. Structural Equation Modelling of the Relationship between Work Stress, *Per3* Gene, and Sleep Quality

Pathway analysis between work stress, the *Per3* gene, and sleep quality based on structural equation modelling was performed on 226 experimental study subjects. The reference evaluation criteria for the structural equation model were $\chi^{2}$/df < 3.0, RMSEA (root-mean-square error of approximation) <0.08, AGFI (adjusted goodness-of-fit index) >0.9, and GFI (goodness-of-fit index) >0.9. The following model was obtained through continuous revision of the model in accordance with the principles of accuracy and simplicity (Figure 1). The model fitted closest to the reference standard among all the models, thus indicating a good fit ($\chi^{2}$/df = 1.716 (41.184/24), RMSEA = 0.056, AGFI = 0.928, GFI = 0.962), and the model was statistically significant (*p* < 0.05).

As shown in Figure 1, in this survey, a direct effect was observed between work stress and sleep quality (*p* < 0.05) with a path coefficient of 0.47, thus suggesting a strong association between work stress and sleep quality among the non-manual workers in Urumqi, Xinjiang. A direct effect was also observed between *Per3* gene polymorphism and sleep quality (*p* < 0.05), with a pathway coefficient of 0.14, thus suggesting an association between *Per3* gene polymorphism and sleep quality among the non-manual workers in Urumqi, Xinjiang; however, no direct effect was observed between *Per3* gene polymorphism and work stress (*p* > 0.05), with a pathway coefficient of −0.07.

## 4. Discussion

The present study aimed to evaluate the sleep quality of non-manual workers in Urumqi, Xinjiang, China, and to analyse the effects of work stress and *Per3* gene polymorphism on sleep quality to further explore the effects of their interaction on sleep. The results of the present study showed that 1038 of 1458 non-manual workers had sleep quality problems, with a detection rate of 71.19%, which shows that the sleep quality problems of non-manual workers in Xinjiang are more serious. Currently, sleep quality problems have become a more common health problem. Sleep quality problems can cause adverse effects such as drowsiness and fatigue, easily trigger psychological problems such as anxiety and depression, and increase the risk of cardiovascular diseases such as hypertension and coronary heart disease and death [29,30,31,32,33]. Therefore, it is particularly important to develop targeted preventive and intervention measures for non-manual workers in Xinjiang to improve their sleep quality.

Previous studies have shown that higher job demands, lower job satisfaction, and work-related psychological factors such as the effort–reward imbalance are associated with the development of sleep quality problems [34]. Several epidemiological studies have shown that work stress is a major risk factor for poor sleep quality [35,36,37]. Work stress is a work-related factor that negatively affects the physical and mental health of people who are exposed to high pressure environments for long periods of time. While moderate stress can motivate workers, excessive stress can have a significant impact on their physical and mental health and quality of life, leading to the suboptimal states of lethargy, drowsiness, and anxiety [38,39]; sleep disorders [40,41,42]; onset of various diseases [43,44]; increased health care and insurance costs; and increased costs for national health services. Currently, a number of domestic and international studies on different occupational groups have shown that work stress directly affects their sleep quality [42,45]. Gao et al. [46] obtained the same results in their study on the sleep quality of doctors. In daily work and life, individuals are prone to sleep problems if they are not relieved of their negative somatic reactions and psychological stress caused by work-related factors such as work stress and burnout. Although work stress does not cause specific occupational diseases, it can affect the physical and mental health of occupational groups, thus increasing the risk of mental disorders, sleep disorders, and cardiovascular disease [47]. Previous studies have also suggested that some alterations in the nervous system, such as dysregulation of the individual autonomic nervous system and the hypothalamic-pituitary-adrenal (HPA) axis, could explain the effects of work stress on poor sleep quality [48]. Kalmbach et al. [49] reported that changes in cortical activity as a response to stress exposure increased the likelihood of future sleep quality problems. Another study [50] suggested that stress is associated with reduced parasympathetic activation, which may have a detrimental effect on voluntary arousal during sleep, which in turn leads to shorter sleep duration.

The circadian biological clock adjusts its circadian profile in response to changes in external environmental stimuli. As a key member of the feedback loop of the biological clock, several studies have reported an association between *Per3* gene polymorphisms and human circadian phenotypes, including early and late night preferences and sleep homeostasis regulation [51,52]. *Per3* VNTR variants are associated with circadian preferences, non-visual responses to light, and brain and cognitive responses to sleep deprivation/circadian rhythm dysregulation. Per3^5/5^ carriers are more likely to go to bed early and wake up earlier than Per3^4/4^ carriers [53,54,55]. Cheng et al. [56] reported that the Per3^5/5^ genotype was associated with an increased risk of daytime sleep disturbance in night shift workers. An association between *Per3* gene polymorphism and sleep quality was also found in a study on sleep quality in Chinese oil workers [57]. Animal studies on this topic have shown that *Per3* protein loss of function in *Per3* knockout mice is associated with altered sleep homeostasis [58,59]. In the current study, regression analysis revealed that the *Per3* gene (*OR* = 3.315, 95% *CI*: 1.672–6.574) was the influencing factor for poor sleep quality. The reason for this finding may be related to differences in the *Per3* VNTR genotype in relation to sleep duration and homeostatic response to sleep loss [20].

The occurrence of sleep quality problems is often influenced by a combination of genetic and environmental factors. In the current study, both work stress and *Per3* gene polymorphisms were found to affect sleep quality in non-manual workers. To further investigate the potential interaction between work stress and *Per3* gene polymorphism in predicting the risk of developing sleep quality problems, the effect of their interaction on sleep was further analysed. Logistic regression analysis showed that, compared to Per3^4/4^× low work stress, Per3^4/5^^,5/5^ × high work stress (*OR* = 2.511, 95% *CI*: 1.635–3.855) had a higher risk of developing sleep quality problems as compared to Per3^4/4^ × low work stress. This finding suggested that an interaction occurs between work stress and *Per3* gene polymorphism and that this interaction increases the risk of developing sleep quality problems. This result is consistent with the findings of a recent study [48]. A study in Greece identified an interaction between *Per3* gene and stressful events in the risk of sleep changes in women [60]. Further pathway analysis between gene–environment–sleep quality based on structural equation modelling revealed a direct association between work stress, *Per3* gene, and sleep quality; however, no direct association was observed between work stress and *Per3* gene, i.e., there was no mediating effect between them. This implies that work stress may not affect sleep quality by affecting the expression of the *Per3* gene. Combining the results of the regression analysis and the pathway analysis, the present study showed an interaction between work stress and the *Per3* gene, but no mediating effect between them.

This study used a cross-sectional approach to investigate the effects of work stress, *Per3* gene polymorphism, and their interaction on sleep quality. Both work stress and *Per3* gene polymorphism were found to affect sleep quality in non-manual workers, and there may be a potential interaction in the risk of sleep quality problems, i.e., there is a cumulative effect of work stress and *Per3* gene on sleep quality, but this interaction may not be achieved through the effect of work stress on the expression of the *Per3* gene. The results of this study have practical implications for improving the sleep quality of non-manual workers in Xinjiang. For individual non-manual workers, it is important to pay attention to the psychological changes caused by their stress in time to prevent sleep quality problems as early as possible; for employers, they should arrange the work tasks of workers reasonably to avoid exposing employees to high work stress environmental stress.

The present study also has some limitations. First, this study relied solely on the PSQI questionnaire to investigate the sleep quality of non-manual workers and was based on respondents’ self-reported sleep, which was too subjective and one-sided. In the future, objective indicators such as polysomnograms could be added to the sleep quality survey to better and more comprehensively analyse the sleep quality of non-manual workers. Second, when investigating the potential mechanisms of gene–environment interactions on sleep, only the effect of *Per3* gene-environment interaction on sleep was studied. In the future, the *Per3* gene and other biological clock genes can be combined to construct gene-gene and gene-environment interactions to investigate the potential mechanisms in the sleep–wake process.

## 5. Conclusions

The current study showed that both work stress and *Per3* gene polymorphism affect sleep quality in non-manual workers in Xinjiang. Moreover, while the findings suggest that there is an interaction between work stress and *Per3* gene polymorphism on sleep quality and that the interaction may increase the risk of poor sleep quality, this effect may not be achieved by a mediating effect. Therefore, avoiding or reducing the stimulation of work-related exposure factors such as high external work stress can effectively reduce the occurrence of sleep quality problems. On the basis of the above results, relevant experts should further develop and improve management systems for non-manual workers and allocate work tasks rationally, and also actively conduct psychological seminars for non-manual workers to provide adequate humanistic care and social support to promote the development of their physical and mental health.

## Figures and Tables

**Figure 1 ijerph-19-06843-f001:**
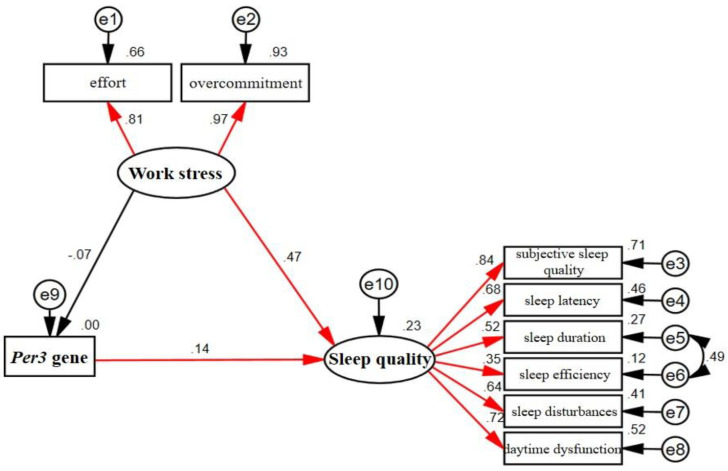
The structural equation model of work stress, *Per3* gene polymorphism, and sleep quality. Note: Red paths indicate statistically significant standardised paths. In the figure, work stress and sleep quality are latent variables, variables that cannot be directly observed and measured, but need to be measured indirectly through the design of several indicators. Thus, effort and overcommitment are explicit variables for work stress; subjective sleep quality, sleep latency, sleep duration, sleep efficiency, sleep disturbances, and daytime dysfunction are explicit variables for sleep quality. Numbers between latent and explicit variables represent factor loading. e1–e10 refer to the residuals of the variable to which the arrow points.

**Table 1 ijerph-19-06843-t001:** Primer sequences.

Gene		Sequence (5′-3′)	Amplified Fragment Length
*Per3*	F	TGT CTT TTC ATG TGC CCT TAC TT	347/401
	R	TGT CTG GCA TTG GAG TTT GA	

**Table 2 ijerph-19-06843-t002:** Genotype information of *Per3*.

Gene	Enzyme Fragment Length	Genotype
*Per3*	347 bp	4/4
	347 bp, 401 bp	4/5
	401 bp	5/5

**Table 3 ijerph-19-06843-t003:** Detection of sleep quality problems in non-manual workers with different demographic characteristics (*n*, %).

Characteristics	Number	Poor Sleep Quality (*n*, %)	$\chi^{2}$	*p*-Value
Gender				
Male	566	398 (70.32)	0.346	0.557
Female	892	640 (71.75)		
Age/(years old)				
<30	456	320 (70.18)	0.772	0.680
30–40	683	485 (71.01)		
>40	319	233 (73.04)		
Education level				
College and below	141	105 (74.47)	0.816	0.366
Undergraduate and above	1317	933 (70.84)		
Occupation				
Teacher	120	84 (70.00)	0.831	0.975
Civil Servants	307	222 (72.31)		
Medical staff	102	70 (68.62)		
Finance, economic operations staff	116	83 (71.55)		
Administrative staff	239	173 (72.38)		
Electrical, construction engineer	574	406 (70.73)		
Professional title				
Elementary	836	608 (72.73)	3.525	0.172
Intermediate	458	311 (67.90)		
Advanced	164	119 (72.56)		
Length of service/(years)				
<5	331	235 (71.00)	3.361	0.339
5~	372	252 (67.74)		
10~	314	230 (73.25)		
≥15	441	321 (72.79)		
Marital status				
Unmarried	480	350 (72.92)	1.036	0.309
Married	978	688 (70.35)		
Monthly income/(yuan)				
≤5000	593	430 (72.51)	0.848	0.357
>5000	865	608 (70.29)		
Smoking				
No	1205	842 (69.88)	5.881	0.015
Yes	253	196 (77.47)		
Alcohol consumption				
No	938	661 (70.47)	0.673	0.412
Yes	520	377 (72.50)		
Total	1458	1038 (71.19)		

**Table 4 ijerph-19-06843-t004:** Condition of sleep quality between different work stress groups (*n*, %).

Work Stress	Number	Non-Poor Sleep Quality	Poor Sleep Quality	$\chi^{2}$	*p*-Value
Low	805	301 (37.39)	504 (62.61)	64.589	<0.001
High	653	119 (18.22)	534 (81.78)		
Total	1458	420 (28.81)	1038 (71.19)		

**Table 5 ijerph-19-06843-t005:** Conditions of PSQI scores between different work stress groups ($\bar{x}\pm s$).

Work Stress	PSQI Scores	Subjective Sleep Quality	Sleep Latency	Sleep Duration	Sleep Efficiency	Sleep Disturbances	Daytime Dysfunction
Low (*n* = 805)	5.68 ± 2.84	1.08 ± 0.69	1.00 ± 0.87	0.77 ± 0.64	0.40 ± 0.75	0.99 ± 0.56	1.45 ± 0.91
High (*n* = 653)	7.24 ± 3.12	1.37 ± 0.75	1.24 ± 0.99	0.99 ± 0.69	0.43 ± 0.78	1.19 ± 0.63	2.01 ± 0.90
*t*	−9.871	−7.719	−4.956	−6.404	−0.579	−6.424	−11.850
*p*-value	<0.001	<0.001	<0.001	<0.001	0.563	<0.001	<0.001

**Table 6 ijerph-19-06843-t006:** Hardy–Weinberg equilibrium test.

Gene	Genotype	Actual Value	Expected Value	*χ^2^*	*p*-Value
*Per3*	4/4	157	158.9	0.871	0.647
	4/5	65	61.2		
	5/5	4	5.9		

**Table 7 ijerph-19-06843-t007:** The distribution of sleep quality across genotypes and alleles of the *Per3* gene.

Gene	Genotype/ Allele	N	Non-Poor Sleep Quality (*n*, %)	Poor Sleep Quality (*n*, %)	$\chi^{2}$	*p*-Value
*Per3*	4/4	157	87 (55.41)	70 (44.59)	6.287	0.043
	4/5	65	24 (36.92)	41 (63.08)		
	5/5	4	2 (50.00)	2 (50.00)		
	4	379	198 (52.24)	181 (47.76)	4.721	0.030
	5	73	28 (38.36)	45 (61.64)		

**Table 8 ijerph-19-06843-t008:** Logistic regression analysis of work stress, *Per3* gene, and poor sleep quality.

Variables	*β*	SE	*Wald*	*p*-Value	*OR* (95% *CI*)
Work stress	1.128	0.323	12.165	<0.001	3.088 (1.639, 5.820)
*Per3* gene	1.199	0.349	11.777	0.001	3.315 (1.672, 6.574)

*OR, odds ratio; CI, confifidence interval.*

**Table 9 ijerph-19-06843-t009:** Interaction between work stress and Per3 gene polymorphism on poor sleep quality.

Comparison Group	*β*	*Wald*	*p*-Value	*OR* (95% *CI*)
Per3 × work stress				
Per3^4/4^ × low work stress	-	-	-	Ref
Per3^4/5^^,5/5^ × high Work stress	0.921	17.705	<0.001	2.511 (1.635–3.855)

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
