# Peer review of "Effects of Work Stress and Period3 Gene Polymorphism and Their Interaction on Sleep Quality of Non-Manual Workers in Xinjiang, China: A Cross-Sectional Study"

_ijerph, 2022, doi:10.3390/ijerph19116843_

Round 1

Reviewer 1 Report

Referee report on “Effects of work stress and Period3 gene polymorphism and their interaction on sleep quality of non-manual workers in Xinjiang, China: A cross-sectional study”

Overall impression: The article investigates the effect of work stress and Per3 gene polymorphism and their interaction on sleep quality among non-manual workers in Xinjiang, China. The topic of the paper is interesting and important, and the focus on non-manual workers is new. The introduction is quite nice and describes the subject sufficiently. However, there are shortcomings. Some parts of the article, especially the empirical analyses, are presented in an unclear way, and need more explaining. The results also need to be verified with adjusted regression models. More specific comments are below.

Specific comments:

Chapter 2.1

Second paragraph: What was the number of respondents after exclusions? I would expect it to be less than 1,458 since you have many exclusion rules. But in the analyses you still have 1,458 observations? Were these restrictions made before the questionnaires were distributed (so these kind of persons are not included in the sample of 1,700)? If so, this need to be clearly stated, at the current form this is unclear.

Line 132. “A total of 113 pairs were successfully matched, resulting in a case-control study of 226 participants.” It is unclear what is the difference between the matched individuals in these 113 pairs? Per3 gene polymorphism or what? This needs more explanation.

Chapter 2.2.1

Line 140. “The questionnaire consists of three dimensions: effort, reward, and overcommitment, with 23 items, of which the first 17 items are scored on a 5-point scale and the last 6 items are scored on a 4-point scale”. How many separate items there are for effort, reward and overcommitment?

It is unclear how the indicator ERI is calculated. “The ERI Index is calculated by assigning equal weightage to each item, with the indicator ERI = (11/6) x (E/R).” Where does 11/6 come from? What does E/R mean in practice? The calculation of ERI needs to be explained more clearly and in more detail.

Chapter 3.1-3.2

Table 4. In general, you cannot conclude that work stress affects sleep quality without regression analysis. In addition to presenting distributions of non-poor sleep quality and poor sleep quality in case of low and high stress, you need to verify the results you obtain in Table 4 with logistic regression and using variables listed in Tables 3 as control variables. You cannot draw conclusions based on just simple Chi2-tests.

The title of Table 4 is not best possible “Condition of sleep quality between different work stress groups”   I think “Association/Connection between work stress and sleep quality” would be better title.  

Chapter 3.3

Line 216. “The PSQI scores on all dimensions differed between the different work stress groups…” I think here it would be good to mention what are the dimensions.

Line 227. I think that your interpretation of the results presented in Table 5 is not correct “…., thus suggesting that the higher the PSQI scores of non-manual workers with high work stress, the worse is their sleep quality” Rather the results suggest that high work stress in connected with lower sleep quality as measured by PSQI score and almost all of its separate dimensions.

The title of Table 5 is unclear “Table 5. Conditions of PSQI scores between different work stress groups” I think “Association/Connection between work stress and the dimensions of sleep quality and PSQI score” would be better title.

It is also unclear what is the number of observations in Table 5.

Chapter 3.5-3.6

Line 240: “The distribution of genotypes and alleles at the 54 bp-VNTR locus of the Per3 gene differed between the sleep quality groups, and the differences were statistically significant (p < 0.05).” I think this is incorrect. As far as I understand, you present the distribution of sleep quality in different Per 3 genotype and allele groups. So the distribution of sleep quality differed between Per3 gene groups, not the other way round.

It is unclear what group of respondents you have used in Table 7. The number of observations is much larger than the number of your matched respondents (226). Where do the rest of the observations (379+79=452) come from?

It is unclear how the logistic regression analysis in Table 7 was performed? What control variables did you use? You cannot make interpretation based on non-adjusted model, so it is necessary to control for other variables.

“Table 7. The distribution of genotype and allele of the Per3 gene in different sleep quality groups and their effects on sleep quality.” As far as I understand, Table 7 does not indicate the distribution of genotype and allele of the Per3 gene in different sleep quality qroups, but it indicates the distribution of sleep quality in different genotype and allele of the Per3 gene groups.

Line 251. In the text you say “The interaction terms for work stress and Per3 gene genotype were further introduced into the multiple regression model.” It is unclear how was the logistic regression analysis in Table 8 performed. What control variables you did use? You need to use the same controls as in previous logistic regression. You cannot make interpretation based on non-adjusted model.

It is unclear what is the number of observations in Table 8? You need to present the number of observations in table 8, in the same way you did in Table 7. So you need to present the distribution of non-poor sleep quality and poor sleep quality with each interaction.

As far as I understand, you have 4 observations with Per3 5/5. Is this really enough for interactions?

Table 8 is unclear. Per3×work stress and Per3 4/4×low work stress are in the same line, as far as I understand, they should not be in the same line. OR is not in the same line with other numbers (B, Wald and p-value) and OR is also not in the same line with the group names (e.g. Per3 4/5×high Work stress).

When it comes to interactions, as far as I understand, you interact Per3 4/4, Per3 4/5 and Per 5/5 with low and high work stress. As far as I understand this makes 6 interactions.

Why in Table 8 you only have 3 interactions? This needs to be explained in more detail.

  • Per3 4/4×low work stress
  • Per3 4/5×high Work stress
  • Per3 5/5×high work stress

What is the number of observations (n) in the structural equation model? You need to present n clearly in Figure 1.

You need to explain in more detail what RMSEA, AGFI and GFI mean.

Figure 1. What do e3-e8 represent? What do e1, e2, e9 and e10 represent? You need to explain the Figure in much more detail.

In Figure 1, what does 0.81 from work stress to effort represent? The same applies to 0.97 from work stress to overcommitment. What do all the arrows from sleep quality into the dimensions of sleep quality, that is, subjective sleep quality, sleep latency, etc. represent? You need to explain the meaning of different arrows in more detail.

Chapter 5.

Line 374. “The current study showed that both work stress and Per3 gene polymorphism affect sleep quality in non-manual workers in Xinjiang….. the findings suggest that there is an interaction between work stress and Per3 gene polymorphism on sleep quality…”

The conclusions are fine on the condition that you obtain the results using adjusted logistic regression models. Without adjusted models you cannot make the conclusion you are making now.  

Author Response

Dear Expert,
Hello! Thank you for taking time out of your busy schedule to check our manuscript. We have revised according to your instructions and uploaded the latest manuscript, please check!

Reviewer 2 Report

A interesting work involving an important sample of subjects for the study of the influence of work stress on the quality of sleep. However, the study of the influence of Period3 gene and its interaction with work stress on the quality of sleep is limited because of the small sample size (113 patients with vs 113 without work stress), and therefore, the results of this interactions should be considered as preliminary. The authors should also discuss on the strengths of their study.

Author Response

Dear Expert,
Hello! Thank you for taking time out of your busy schedule to check our manuscript. We have revised according to your instructions and uploaded the latest manuscript, please check! (see P.12 Line389-394)

Round 2

Reviewer 1 Report

Referee report on: “Effects of work stress and Period3 gene polymorphism and their interaction on sleep quality of non-manual workers in Xinjiang, China: A cross-sectional study”

The authors have responded to most of my concerns and questions, but there still is a couple of details that are not clear to me or that need clarification.

Line 145 ” The effort- reward imbalance index is calculated as: the score of effort / (the score of effort x C)” Is this really so? There is no score of reward in this formula?

Line 227: “3.3. Association/Connection between work stress and the dimensions of sleep quality and PSQI score” You need to select either association or connection, not use both in the title. Association and connection were just alternative ways of saying the same thing. I suggest you use “Association” as in the previous title (3.2).

Line 248 “Table 7. The distribution of genotype and allele of the Per3 gene in different sleep quality groups and their effects on sleep quality” The title is still incorrect, I think the title should be “Distribution of sleep quality across genotypes and alleles of the Per3” because you present the distribution of sleep quality, not the distribution of genotype and allele of the Per3 gene.

Line 251 “Of the 226 experimental study subjects, the sleep quality status served as the dependent variable, while work stress and Per3 gene genotypes were considered to be independent variables.” I think you need to say clearly that you use poor sleep quality vs. non-poor sleep quality as dependent variable.

Line 256 “…were influencing factors for sleep quality after adjusting for confounding factors, such as occupation, length of service, education, and monthly income.“ You need to mention poor sleep quality, so “…..were influencing factors for poor sleep quality…”

“Table 8. Logistic regression analysis of work stress, Per3 gene, and sleep quality” You also need to mention poor sleep quality in the title of table 8

“Table 9. Interaction between work stress and Per3 gene polymorphism on sleep quality” You need to mention poor sleep quality in the title of table 9.

Table 9: In the table, there is nothing under the heading “Reference group”, so what is the purpose of this heading?

Author Response

Thank you for your comments, please see the attachment for the corresponding response.
